# Price Discrimination in the Online Airline Market: An Empirical Study

**Stefano Azzolina [1], Manuel Razza [2,†], Kevin Sartiano [3] and Emanuel Weitschek [2,3,*,†]**

1   Department of Economics, University of Bologna, Piazza Scaravilli 2, 40126 Bologna, Italy; stefano.azzolina2@unibo.it
2   Italian Competition Authority, Piazza Verdi 6/a, 00198 Rome, Italy; manuel.razza@agcm.it
3   Department of Engineering, Uninettuno University, Corso Vittorio Emanuele II 139, 00186 Rome, Italy; k.sartiano@students.uninettunouniversity.net
*   Correspondence: emanuel.weitschek@uninettunouniversity.net
†   The views and findings expressed in this article are those of the authors and do not represent the opinion or policy of their institutions. Examples of analysis performed within this article are only examples. Assumptions made within the analysis and the results are not reflective of the position of the Italian Competition Authority.

**Abstract:** Big Data technologies have significantly increased the possibility for sellers to adopt personalisation strategies, especially in digital markets. Among such strategies, price discrimination, a practice where the same commodity is sold at different prices, either to the same customer or to different customers, stands out. Particularly, the online airline ticket market has risen the attention of economists in recent studies, both because of its specificity and of the high data availability. This manuscript enters the debate and analyses the airline ticket market in an original way. Indeed, the aim of this work is to empirically understand whether some airline companies discriminate in prices by using the customers' data that they collect from their websites, and, if so, which is the impact on social welfare. For performing this assessment, a software that is able to automatically collect pricing data is developed and freely released. By executing the software in two time periods, tickets prices of three airlines, three itineraries, and four different user profiles are acquired. A double analysis is performed to check if customers' information are used to discriminate (intra-user), and if different prices are offered to distinct user profiles (inter-user). Moreover, the analyses consider control data collected from the API of the Global Distribution System Amadeus, the main flight booking platform dedicated to travel agents. Upon inspection, no evidence is found in this study to support the hypothesis that airlines use price discrimination techniques.

**Keywords:** price discrimination; digital markets; airlines; pricing; personalisation

## 1. Introduction

Thanks to the advancements of digital and big data technologies, especially in the economic and social systems, new opportunities for existing business strategies are emerging. In industry and commerce, particularly relevant is the possibility for the supply side to acquire much more detailed information about the demand side. Indeed, the producer or the seller has now the ability to better understand the needs, the preferences, and the financial situation of customers so as to reduce asymmetric information and finely tailor the offer. For this reason, economists and research scholars are devoting particular attention to monitor the effects on pricing strategies, since granular information could foster personalisation strategies, like price discrimination. Discriminating in prices means charging different prices according to the consumers' willingness to pay (WTP). When the data is used to exploit customers' WTP, this could result in negative price discrimination.

In recent years, price discrimination has come under increased investigation in the airline industry, i.e., airlines have been suspected of using user data to exploit the customers'

WTP. The aim of this study is to empirically assess whether and under which conditions some airlines use price discrimination based on the personal information they collect from their customers on their websites Indeed, within their direct booking channels airline companies are able to collect several personal information from their customers such as the operating system, the browser, the geographic location.

In order to assess the presence of price discrimination, we design and develop a software that is able to automatically search and acquire pricing data from the websites of the airlines. Additionally, the software exploits the API of the Global Distribution System Amadeus, the main flight booking platform dedicated to travel agents, in order to retrieve control data. Indeed, our research distinguishes itself from previous works, as it looks not only at the price difference offered to customers, but also at the fare bases. It is worth noting that, we release the software publicly and by adopting the open-source paradigm to enable other researchers to perform further research on different airlines, itineraries, and time frames.

Over the course of six weeks, we acquire each day at four different time stamps pricing data of three different airlines (i.e., Alitalia, Ryanair, and Lufthansa) for one national and for one continental itinerary to determine whether price discrimination occurs. Two acquisition are performed, the first one long time ahead the dates of the flights, the second one near the departure. Four user profiles are identified to check whether there is a link between the device brand and type used to execute the search and the offered price. These profiles take into account the most common operating systems and browsers available today. The flight data obtained from the websites are also compared to a control group in order establish if the observed airlines use price discrimination when they have access to customers' information. Specifically, we analyse the collected data by performing two types of inspection: intra-user profile and inter-user profile. The first one to ensure that no difference occurred among prices offered to the same user at different times with and without the presence of cookies. The second one to focus on the prices offered to each user profile defined by the operating system and browser. In a nutshell, by inspecting these data we want to investigate the direct relationship between the travelers and the airline companies so as to either confirm or reject the results that have already been obtained in studying this market: do seller change price according to the device through which one search for the ticket? Is there any browser-related price differentiation? If a difference in price exists, should it be considered as the result of the huge volatility of prices characterizing this industry? Or is it the effect of a specific strategy? Upon inspection, no evidence is found in this study to support the hypothesis that airlines use price discrimination against their customers. This holds valid for both intra- and inter-user analyses.

The rest of this manuscript is organized as follows. Section 2 Background describes the fundamentals of our work by elaborating the concept of price discrimination (Section 2.1), by summarizing the results presented in the already existing literature (Section 2.2), and by describing the structure of the airline ticket market (Section 2.3). Section 3 Materials and Methods illustrates the empirical investigation we have conducted. Precisely, it analyses the experimental setup and the tools we have adopted. Additionally, in Section 3.2, it briefly accounts for the architecture of the software we have built. Further information about the software and about the tools we have applied are provided in Additional File S1 (Supplementary Materials). Section 4 Results and Discussion explores the data, analyses the results, and illustrates the economic implications. Finally, Section 5 Conclusions and Future Directions summarizes the main facts of this work and paves the way for future researches.

## 2. Background

In this section we provide the reader with a comprehensive definition of price discrimination by describing the economic theories and by illustrating several state-of-the-art previous studies. Additionally, we introduce the airline tickets market by explaining its peculiarities.

### 2.1. Price Discrimination

According to J. Tirole [1], price discrimination occurs whenever the same commodity is sold at different prices, either to the same customer or to different customers. More precisely, G. Stigler [2] states that price discrimination arises when two or more similar goods are sold at prices that are in different ratios to marginal costs. The latter definition rules out the differences in prices determined by the differences in costs of serving different customers. Moreover, it includes the case in which the seller discriminates by setting a uniform price.

The conventional economic theory adopts the classification made by [3] and distinguishes among three different categories of price discrimination. Adopting *first-degree price discrimination* (a.k.a. *personalized pricing*), the seller is able to set a price equal to the willingness-to-pay (WTP) of each consumer so to extrapolate the entire social surplus. This strategy has often been considered as a useful but abstract benchmark. However, big data and the behavior-based price discrimination are rewriting this wisdom: as highlighted in [4], first-degree price discrimination is nowadays a more realistic framework since each single user produces enough personal data for the seller to infer about one's tastes and possibilities. Similarly, *third-degree price discrimination* (a.k.a. *grouping*) occurs when different prices are assigned to different groups of customers. Each group is composed by customers who share some common features (e.g., business vs. leisure travelers). In other words, this kind of discrimination is a less granular form of first-degree price discrimination: that is why the more modern categorization addresses both of them as *direct price discrimination*. Finally, the remaining category is the *second-degree price discrimination* (a.k.a. *versioning*), that is the *indirect price discrimination*, and it is based on indirect signals. In this case, the seller lacks of information, and so sets a menu of prices and offers among which the consumer can choose. By properly designing these bundles, consumers will have an incentive to truthfully reveal their preferences and WTPs (see [5]). The offers may differ either in quantity or in quality. In line with [6], several kind of indirect price discrimination strategies may be identified: coupons, quantity discounts, bundling, performance-based discrimination, restrictions on purchase and use, knowledge-based discrimination, and non-linear pricing.

It is worth noting that the advent of data analysis and the development of behavioral marketing have triggered the rise of a new category of price discrimination strategies: the *behavior-based price discrimination (BBPD)*. According to [7]: *sellers are now using big data and digital technology to explore consumer demand, to steer consumers towards particular products, to create targeted advertising and marketing offers, and in a more limited and experimental fashion, to set personalised prices. At the same time, buyers are making use of the Internet and the variety of choices and tools it provides to ensure that they get a good deal*. For sure, BBPD can be defined as a more sophisticated kind of direct and interpersonal price discrimination. However, it can also be seen as a mix of strategies whose aim is to indirectly incentivize the consumer to autonomously select her type. Economic scholars have widely scrutinized these practices. Among them, *add-on pricing scheme* and price obfuscation strategies (see [8]), price inertia (see [9]), and price dispersion (see [10]) are widely adopted in online flight booking too. All these strategies can be traced back to the behavioral biases that affects human behavior, like information overload, status quo bias, loss aversion, framing, hyperbolic discounting. A good summary of the ones which behavioral marketing usually refers to is included in [11].

Before going on, it is important to distinguish personalized from dynamic pricing. The only common feature is the use of new technologies and the goal, i.e., to maximize firm's

profit. This net distinction has been very well studied in all its facets by [12]. Dynamic pricing is that practice through which online seller use algorithms to constantly update their prices in line with the information related to both the demand and the supply side of a market. Since the information that sellers can get online are abundant and it is becoming increasingly cheaper, then it is quite common to elaborate these data to infer on the market conditions in order to choose a price that can more likely maximize producers' welfare. Anyway, dynamic pricing does not involve any kind of discrimination, since it is not related to a specific target of consumers but concerns the whole market and its equilibrium.

This distinction is crucial when one studies a market such as the online flight booking one. Indeed, ticket prices are constantly updated according to many factors such as seat availability, season, country of purchase, length of stay, and so on. These variables make ticket prices quite volatile and dynamic.

There are three major underlying conditions whose absence would prevent the seller from personalizing prices [13]. First of all, the seller must have the opportunity to set a price, i.e., she must have some market power. If no market power is in the hand of the seller, then there is no chance to make any decision about the price of any product. Second, a *no arbitrage* condition must holds. Indeed, it is necessary to prevent buyers from generating secondary markets in which the customer whose WTP is low resells the commodity to one whit a higher WTP. Indeed, under price discrimination the low type—i.e., the buyer with a low WTP—pays a lower price than the one paid by a high type—i.e., the buyer with a high WTP- and this opens to secondary markets. If this *transferability of the commodity* is feasible, then all the effort of the seller would be useless and this strategy would not be profitable anymore. Third, it has to be possible to obtain information about the customers and, more in general, about the structure of the demand side. If not, it would be nearly impossible both to evaluate differences among consumers and to segment the market.

For what concerns the online airline market, it is reasonable to expect some price discrimination to arise. As it will be better explained later, this market is an oligopolistic one, where firms have some market power. Moreover, they are able to collect information about the demand side. Filling this gap has been considered an important part of this business since the birth of the airline market, and the ability in gathering information has sharply increased with the advent of the digital economy. Furthermore, flight tickets are very difficult to resell because of their specificities, and the increasing personalization makes this practice even harder.

Looking at the possible effects, in general price discrimination is perceived as a welfare-enhancing strategy, and so regulatory agencies tend not to ex-ante prevent it.

As explained in [14], a major role is played by the *market expansion effect*. By price discriminating, the producer increases the amount of output, and sets up a price schedule that makes that good affordable also for previously excluded customers. As a second positive side, personalized prices lower the search cost that the customers should normally face (see [10]). Moreover, if one considers a non-monopolistic setting, then price discrimination strategies may strengthen competition. As mentioned in [15], the possibility to address customers with targeted discounts may reduce the barriers to entry. At the same time, price discrimination can be read as a segmentation of the market that generates a conspicuous amount of smaller and less heterogeneous niches of customers, and it is possible that the original barriers to entry are not strong enough to prevent the entrance of a new competitor in a single fraction of the original market.

However, as described in [16], there might be cases in which price discrimination may harm social welfare and especially consumer surplus. These cases are much more likely to arise in (quasi-)monopolistic markets [15]. The most evident negative side is the so-called *appropriation effect*: contrary to the market expansion effect, this is that exploitative practice through which the seller offers a price higher than the uniform one to those already-in-the-market customers whose WTP is high enough (e.g., loyal customers). In this way, the seller is able either to counterbalance the rise in costs due to the increase in the size of the output or simply to extract some rent from the clients. Moreover, price discrimination

may be harmful when it is not transparent. In this case, consumers' behavioral biases may be leveraged to artificially shift up the naive customers' demand. At the same time, the lack of transparency may incentive the breach of the rules of fairness, such as the prohibition of discrimination based on gender, race, or disability (see, for example, [17,18]). In general, the use of this practice may reduce the level of trust inside the market. Finally, considering a non-monopolistic market, personalized pricing may be harmful whenever it is adopted by a super-dominant firm as an exclusionary practice to force competitors out of the market (e.g., predatory pricing, fidelity rebates, loyalty discounts, bundled discounts and rebates, margin squeeze). In this situation, the cost to discriminate may be too high for the competitors and the beneficial effect may be too low to balance the loss of competition.

### 2.2. Previous Studies

When analysing the economic literature on airline ticket pricing, the first important result to highlight is the huge volatility. The impact of several dynamic factors—flight distance, fuel price, air traffic, flight classes, seasonality—generate fares which differ even for seats on the same flight. Therefore, a first strand is devoted to the study of dynamic pricing. However, addressing this phenomenon as discriminatory may be a mistake. Indeed, in most cases the change in price is simply the result of an update of the available market-related information. Nevertheless, evidence has been found about what has been defined as *dynamic price discrimination* [19]. According to the previous definitions, it can be seen as a mix between grouping and product differentiation. The typical example is the trend of the airline companies to offer higher prices during office hours and lower prices in the evening. This scheme is a screening mechanism through which the consumers have incentive in autonomously reveal their types: business travelers tend to buy while they are at work and they have a low elasticity of demand, while leisure travelers usually buy a ticket in the evening and they are characterized by a more elastic demand. As a consequence, economic scholars have focused their attention on the perfect timing theory to buy a plane ticket. Among the others, the authors of [20] analyse Russian airline ticket market and compares local and global flights price behavior for the spring-summer 2015 period in the two main hubs in Russia (i.e., Moscow and Saint Petersburg) selecting 50 most popular destinations from Moscow. For each day a request to get the minimum price has been done. In line with the results obtained in previous works, they conclude that it is better to buy either in advance to prevent price increases in the future or few days before departure. However, this result is not valid for internal flights, which are highly influenced by the lack of competition and the absence of low-cost carriers. Similarly, the authors of [21] testifies how the lack of information on the companies' fares makes difficult for the buyer to determine the perfect purchase time, even when historical data series are available. By collecting consumers-available data and using a lag scheme to include lagged features, the authors build a PLS regression model to predict prices. The results obtained through the experiment show that buying as early as possible is not always the best policy. For example, airlines can change fares until the last moment, lowering prices either to increase sales or to fill unsold seats. Therefore, committing to a specific ticket a long time before the flight may be risky, and the use of consumer algorithms may be cost-saving and beneficial. In line with this reasoning, the authors of [22] built a forecasting system to help consumers in purchasing tickets by combining an ARMA algorithm and a random forest algorithm. By using data from nine cities in China and taking into account crucial variables such as take-off times, departure date, and competition from other airlines, they demonstrate how this model can be effective in predicting future prices. Another interesting forecasting model has been developed in [23] by means of Machine Learning techniques. Once collected data about twenty flights between the 5 major American hubs (Atlanta, Chicago, Los Angeles, Dallas, Denver) for one week, they run a model to predict the price at a future date, the minimum value of a fare, and the expected fluctuations of the price. In conclusion, it is necessary to remind that the big flaw of these systems is the lack of data: even if the information of the average ticket price can be extracted from

travel sites, the one regarding the prices of specific flights or the number of available seats on the flight are not always made public either for reasons of competition or because of private negotiations.

Apart from this first block of research, a second part of the literature focuses on the search for other and more classic forms of price discrimination in the digital markets. Our paper enters this debate. The leading works about online price discrimination are [24–27] since they found the first empirical results by using pioneering empirical methods. Once collected the data, these researchers try to measure price discrimination by ruling out any possible source of noise. Indeed, two prices assigned to the same product may diverge for reasons other than the difference in the willingness to pay (WTP): technical factors such as the distributed infrastructure or the update of the search index may be a first explanation, as well as A/B testing and the differences in geolocation that, in turn, determines differences in monetary conversion, taxation, and shipping costs. Looking at the market we are studying, an interesting phenomenon is the *caching of prices*: for cost reasons indeed user data is stored in the cache, which records the essential information of a site to use it faster if needed. This can cause a spread between the price that buyer sees the first time on the site and final price when purchasing the ticket. Moreover, one should also be able to disentangle personalized prices from dynamic price discrimination. Once the noises are silenced, any remaining difference has to be attributed to a different WTP. However, in most cases price discrimination is absent. Apart from the famous case of Amazon's DVDs in 2000 and interesting journalistic investigations ([28,29]), several studies have demonstrated that online price discrimination exists even if it is rare and hard to measure (see [30–32]). Among the others, in 2011 TheTrainline.com, Expedia, Easyjet, Virgin, Lastminute and Eurostar were accused of price discrimination based on previous research or queries. By contrast, strong empirical evidence about *search discrimination* have been found. For example, Ref. [28] reported that Orbiz, an online travel agency (OTA) first offered Apple laptop owners the most expensive hotel and travel results. Similar works have focused the attention to the online airline ticket market, since the advent of the new technologies has had an important impact on this market (e.g., [33]), as it will be shown in the subsequent paragraph. Particularly, Ref. [34] is the one that more closely resembles the approach we adopt. It emulated the practice of searching for users for a specific flight. In doing so, they run a three-week experiment, analysing 25 companies with a dozen profiles (with three different user's profiles: affluent, budget and flight comparer), deactivating or activating the tracking systems (cookies) in different browsers (Chrome, Safari, Internet Explorer) and in different geolocations (using two IPs), and they produced 130,000 queries. This type of research was carried out twice a day for each Airlines in a consecutive and non-simultaneous fashion so to avoid blocking the airline's website server. In line with the more general results, no evidence was found of price discrimination phenomena between the different profiles.

### 2.3. Airline Tickets Market

Before introducing the empirical analysis, it is important to briefly summarize the structure of the airline tickets markets to fully understand the subsequent steps. As shown in Figure 1, apart from travelers (i.e., the demand side), the airline tickets market mainly consists of: airline companies (e.g., Alitalia, Rome, Italy; Lufthansa, Cologne, Germany), Global Distribution Systems (GDSs) (e.g., Amadeus, Madrid, Spain; Sabre, Southlake, TX, USA; Travelport, Langley, Slough,United Kingdom), online travel agencies (OTAs) (e.g., Viator, San Francisco, CA, USA; Expedia, Redmond, WA, USA), content aggregators (e.g., Skyscanner, Edinburgh, Scotland, UK; Kayak, Stamford, CT, USA), and consolidators (e.g., Mondee Group, Silicon Valley, CA, USA). They tend to adopt common standards and protocols to share information and regulate the market. Traditionally, the airline companies use to set prices and tariffs according to the flights information, and then to transfer these offers to the Global Distribution System (GDS). These few players, in turn, use their network to transmit these offers to the travel agents. Among the remaining

players, the GDSs usually interact with OTAs, which offer customers the possibility to comprehensively organize their travel, from flights booking to hotels. Differently than OTAs, the content aggregators are search engines that help customers in comparing all the available offers and then redirect them to the service provider. Finally, the consolidators are middlemen that sign private agreements on tickets tariffs with the airline companies, and resell them to large companies, or OTAs. Figure 1 provides a schematic overview of the ticket distribution channels.

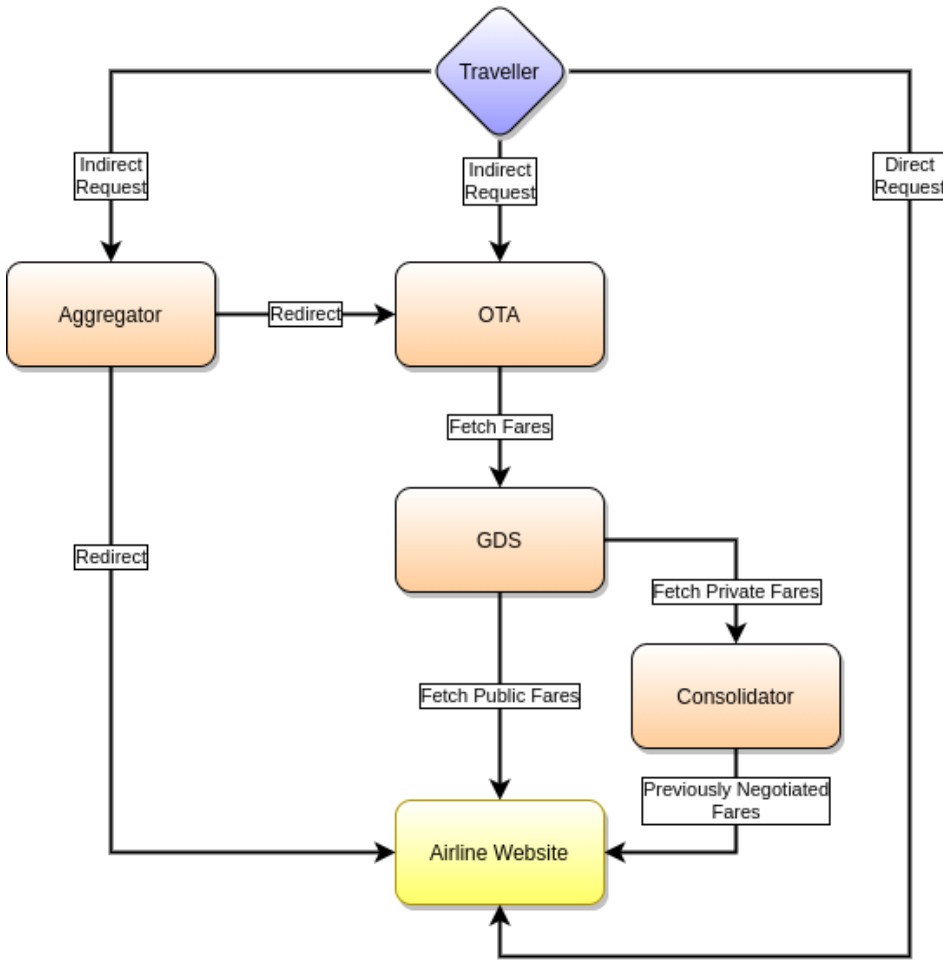

**Figure 1.** Ticket Distribution Channels.

However, the advent of big data and AI is remarkably reshaping this market. Because of its characteristics, the aviation sector was one of the first to make investments in analytical and intelligent technologies that include data analysis, dynamic dashboards, classification and targeting techniques to increase efficiency in airport management and to optimize costs. The empirical analysis conducted by the authors of [35] states that airlines need to adopt a data mining systems due to the huge amount of data generated. This study proves how the data infrastructure and series analysis help airlines in sales and logistics management. Indeed, airlines that adopt data analytics are able to offer better sale conditions to consumers, to change the prices dynamically and to avoid unsold seats. The second aspect concerns marketing: data mining allows to customize customers' travel experience, manage delays and reschedule reservations. In other words, airlines have the possibility to both differentiate their products and better personalize customers' experience and offers by including discounts and ancillaries. This returns positive effects to the companies in terms of brand loyalty, customer satisfaction and increase in revenues. The third aspect is related to the improvement of airport management: data analysis tools helps to analyse passenger flows, the time spent between arrival and take-off. It makes possible to

maintain a high-quality standard, which is also useful for increasing safety systems. The fourth aspect concerns the efficiency in terms of costs: intelligence tools enable companies to analyse the atmospheric conditions or flight speed, reducing times and fuel costs. Finally, having access to historical data improves the booking process and the proposal of accessory services (favorite places, hotels, and transportation). In order to fully exploit the use of big data and machine learning (ML), in 2012 the International Air Transport Association (IATA) has replaced the old EDIFACT protocol with a new API standard, the New Distributed Capability (NDC). The aim is to allow the airline companies to leverage the possibility to directly bargain online with customers in order to gather previously-inaccessible granular information. In doing so, they basically bypass the content aggregators scaling back the dominant role they have had. More precisely, as already experienced by OTAs, airline companies may benefit from the adoption of tracking mechanisms. As explained in [36], online sellers may collect data in infinitely many ways: there are session-only tracking mechanisms, but also storage-based, cache-based, supercookies, fingerprinting. Browser cookies allow a web server to store a small amount of data on the devices of visiting users, which is then sent back to the web server upon subsequent request. Moreover, in line with [37], it is possible to track by means of first-party cookies or third-party ones. While the former are used transparently by online platforms to recognize users, the latter are those that come from third parties, they are more invasive and provide content external to the first-party page. By using these tools, online platforms may personalize search paths and results, recording user information and exploit targeted advertising or selling the information to other platforms. An online platforms in fact could recognize the user profile over time (through the use of cookies) and gradually increase the selling price, exploiting the fear that the price could rise again and encourage the user to purchase.

In conclusion, it is worth noting that the implementation of the NDC is not easy at it may seem: while airlines are relying on IT firms to build the proper original NDC, the GDSs are adopting the new standard too, and they are still maintaining their strong position (for a more detailed exposition, see for example [38] or [39].

## 3. Materials and Methods

The goal of this research is to examine whether some airlines apply price discrimination based on the individual data that they collect from the customers through their direct booking channels, i.e., their websites. In order to empirically test such discrimination, we collect data four times a day from searches about itineraries of three different airlines performed by distinct user profiles. We compare the prices of the searches and assess if discrimination occurs. Two data acquisitions experiments have been performed, the first one lasted four weeks and was performed starting from five months before departure dates, the second acquisition lasted for two weeks and was performed starting from three weeks before departure dates. A variety of factors come into play when the retail price of an airline ticket is determined, such as the type of travel or the time of the booking. A finite number of search parameters has been therefore selected. For every parameter (itinerary, fare type, number of passengers), the focus was placed to determine whether the observed airline company uses price discrimination against their customers. As depicted in Figure 1, several third-party companies are included when a user searches for a flight through an indirect channel. If a price difference among user profiles was found in this case, it would be impossible to identify whether this was due to price discrimination applied by the airline company or by the other involved parties. In addition, it would be impossible to tell whether the offered fare was a published or private one. Private fares pose a risk in the price analysis, because they would always be cheaper than the respective published fare available. Hence, when comparing two prices with each other, there would be no certainty whether an observed price difference was due to price discrimination or to a special agreement between the airline and the third party. To avoid these limitations, only the direct booking channel was considered, in which the customer searches for a flight on the website of an airline and can only be offered public fares.

In the following we describe the experimental setting, the developed software, and the obtained results.

*3.1. Experimental Settings*

The first data collection was performed in the span of one month starting from five months before departure dates of the flights, i.e., 16 July and 23 July 2021. The second acquisition lasted for two weeks and was performed starting from three weeks before departure dates. Therefore we checked for price discrimination long time before and near the flights' dates. Each day, the same search was completed at the exact same four time stamps. The actual hours were picked randomly, however the goal was to make sure that the three parts of the day (morning, noon, evening) were represented as users are expected to typically look for flights around those times of the day. The early search was done at 9:00 a.m. and the later one at 2:00 p.m. In the evening, two searches were done: one at 6:00 p.m. and the other one at 6:30 p.m. These times were picked in order to determine whether there is any price difference between searches that are separated by several hours (9:00 a.m. and 2:00 p.m.) versus searches that are completed within a short time span (6:00 p.m. and 6:30 p.m.).

Two itineraries have been selected to represent two different types of routes, one national, i.e., Rome (FCO)—Catania (CTA), and one continental route, i.e., Rome (FCO)—Munich (MUC). Three airlines have been identified to carry out the data collection, each one to represent a specific category: one national air carrier, i.e., Alitalia (AZ); one low-cost carrier, i.e., Ryanair (FR), and one continental air carrier, i.e., Lufthansa (LH). Alitalia and Ryanair were used to sample data on the national route, whereas for the continental route Lufthansa flights were analysed. For all three carriers, a sample itinerary was defined as follows: a single passenger is travelling for a weekend in July carrying only a hand-luggage. The details of all three itineraries can be found Table 1.

**Table 1.** Itineraries of the flights.

|                      | **Alitalia**       | **Ryanair**        | **Lufthansa**      |
| -------------------- | ------------------ | ------------------ | ------------------ |
| Departure Airport    | Rome (FCO)         | Rome (FCO)         | Rome (FCO)         |
| Departure Date       | 16 July 2021       | 16 July 2021       | 23 July 2021       |
| Departure Time       | 17:00              | 17:50              | 19:15              |
| Departure Flight     | AZ1733             | FR4872             | LH1871             |
| Return Airport       | Catania (CTA)      | Catania (CTA)      | Munich (MUC)       |
| Return Date          | 18 July 2021       | 18 July 2021       | 25 July 2021       |
| Return Time          | 20:20              | 20:10              | 16:55              |
| Return Flight        | AZ1752             | FR1160             | LH1870             |
| Number of Passengers | 1 Adult            | 1 Adult            | 1 Adult            |
| Fare Brand           | Hand-luggage fare  | Hand-luggage fare  | Hand-luggage fare  |

It is a regular practice for airline companies to change their flight schedules. Sometimes they are obliged to do so because airports must guarantee a safe level of air traffic. Other times companies choose to make such changes in order to meet internal operational needs. During the one-month span of the research, this has happened once: both Lufthansa flights that were originally chosen were affected by a schedule change. The departure flight LH1871 has suffered from a time change and the return flight LH1870 has been cancelled. In order to continue the data collection, the flight with the most similar characteristics (LH1872) was selected.

To determine if price discrimination occurs based on the type of device from which the search is performed, four user profiles were defined. These profiles take into account some major operating systems and browsers on the market right now. To ensure that every single search was performed from the same geographical location, for each profile the search was performed using a persistent IP address. This was made possible by using a Virtual Private Network (VPN) service. The different user defined in this study including

their operating system, type of device (mobile or desktop), and IP address are summarized in Table 2.

**Table 2.** Considered user profiles.

| User | Windows-Chrome | Android-Chrome | Macos-Safari | IOS-Safari |
|---|---|---|---|---|
| Operating System | Windows 10 | Android 10 | macOS 10.15 | iOS 14.3 |
| Developer | Microsoft | Google | Apple | Apple |
| Device | Desktop | Mobile | Desktop | Mobile |
| Browser | Chrome 87 | Chrome 87 | Safari 14.0 | Safari 14.0 |
| IP Address | 185.183.105.28 | 82.102.21.68 | 192.145.127.236 | 37.120.201.244 |

In order to determine whether the observed airlines make use of price discrimination, the flight data that is collected from their websites have to be compared with a control group. This is achieved by retrieving information about the same flights directly from the GDS. As presented by [40], one of the main limitations of the indirect distribution channels based on GDS, is that airlines cannot personalize their products and services based on customers profile and history. GDSs rely on the outdated EDIFACT communication standard and this only allows the exchange of limited passenger information, which is insufficient for passenger identification. If the airline cannot identify the individual user, it is impossible to apply price discrimination. Therefore, the data retrieved from the GDS can serve adequately for control purposes.

Historically, GDSs have provided functionality through terminal applications. In recent years, the major GDS providers started to offer APIs to allow access to their functionalities to any authorised developer. For this research, the *Amadeus Self-Service APIs* have been chosen to collect the control group data. In particular, the *Flight Offers Search* endpoint has been selected to perform the searches and retrieve flight information for Alitalia and Lufthansa. This endpoint allows to send a request specifying the itineraries presented in Table 1 and receive the requested flight details which include a variety of information such as prices, fare details, airline names, baggage allowances and departure terminals. From these, the price, fare basis and availability have been extracted to be compared with the data obtained from the airline websites.

Amadeus API Limitations: Lufthansa Fare

When searching for Lufthansa flights with the Amadeus APIs, there are a few key factors that need to be taken into account. Lufthansa has in place a Distribution Cost Charge (DCC) of €19.00 for each booking done via GDSs to encourage travel agents to create bookings via NDC channels. Another aspect to consider is that, even by requesting a fare which includes only a hand-luggage (Lufthansa Light Fare, LGT), every time the retrieved offer comprises a checked baggage with an extra cost of €40.00 (Lufthansa Classic Fare, CLS). Finally, there is always a €20.00 fare surcharge which can be observed by the recurring presence of a *P* in the second-last letter of the presented fare basis. The nature of this is unclear, but it is a constant of every search performed and it is believed to be linked to the use of the APIs. All limitations considered, for each Lufthansa search, a total of €79.00 have been deducted from the control data to maintain consistency throughout the research.

Amadeus API Limitations: Ryanair Search

Due to the fact that Ryanair and Amadeus do not have a commercial agreement, as shown in [41], the control data for Ryanair could not be collected using the Amadeus APIs. Instead, a cookie-less search has been used. As a result of this limitation, it was not possible to collect fare basis information for Ryanair.

Elements to Asses Price Discrimination

The first indicator that could suggest that there is price discrimination is a difference in price. This condition, although necessary, is not sufficient to prove price discrimination as there are also other factors that can cause a price difference. A case in point is when a fare class is sold out and there is a significant jump in price to the next available fare class. The second indicator of price discrimination is the fare basis. If two users are shown two different fare bases, this means that they are fundamentally provided with two different offers with regards to the fare conditions. This would result in a price difference but as a consequence of offer discrimination. To further explain the subtle difference between price and offer discrimination, an exemplary case is presented. Two users make the same search, one using a macOS laptop and the other one using a Windows laptop. The macOS user, who is believed to be willing to pay more, is offered a flight comprehensive of a checked baggage (corresponding to the fare basis KEUCLSX3). The Windows user, who is believed to be more price sensitive, is offered a flight including only a hand-luggage (corresponding to the fare basis KEULGTX3). In this case the discrimination occurs on the offer rather than the price. Once again, a fare difference is not sufficient to prove discrimination as a lack of availability in the requested fare class would also result in different offers. As previously mentioned, the availability must be monitored to exclude false-positives from the analysis. During the data collection the maximum number of simultaneous searches for the same flight was two. For this reason, if a price or fare difference is observed, in order to infer discrimination, a minimum of two seats must be available.

*3.2. The Flight Data Acquisition Software*

In order to perform data collection we designed and developed a new software that we release publicly on Github and that is available at https://github.com/kevinsartiano/airline-price-discrimination-project, accessed on 6 September 2021. The software has been written using the Python 3 programming language and it was designed with the idea of scalability in mind. The software supports data collection from three air carriers. To facilitate the addition of other air carriers in the future, it was important to understand the similarities and the differences of the flight search workflow.

Flight Search Similarities

At first, on the carrier website, the user is presented with an interactive widget that allows the selection of the flight search parameters. These include departure and arrival airport, number of passengers, cabin of preference and type of itinerary (one-way or round trip). Upon confirmation, the website uses the selected parameters to execute a script on the server-side. This phase corresponds to the *loading search results* page. The user is then presented another interactive widget with the list of search results and available offers. After having selected the preferred time of departure and fare brand, the final offer is shown to the user. It can be concluded that the flight search process is similar among the airlines. The steps involved in the process are *flight search*, *offer selection* and *offer confirmation*.

Flight Search Differences

The way an airline decides to implement the selection widgets varies. In some cases the airport and date selection can be done by sending a simple string of text. In other cases, the preferences can only be expressed by clicking on context menus with predefined options. In addition, the HTML layout as well as the class and the attributes of the presented elements vary significantly. Therefore, the way the scraper interacts with each HTML element requires a dedicated handling.

After having analysed the flight search workflow, the *template method design pattern* was chosen to design the scraper. The main reason for choosing this pattern is that it allows scalability to extend to further airline carriers for future researches. The architecture of the software is shown in Figure 2. The software consists of four main components, i.e.,

*the scraper*, which handles data acquisition, *the airline carrier* that models an airline, *the user profile* that represents a customer with its browser and operating system, and *the flight itineraries*. For more details about the software architecture and its modules the reader may refer to Additional File S1 (Supplementary Materials).

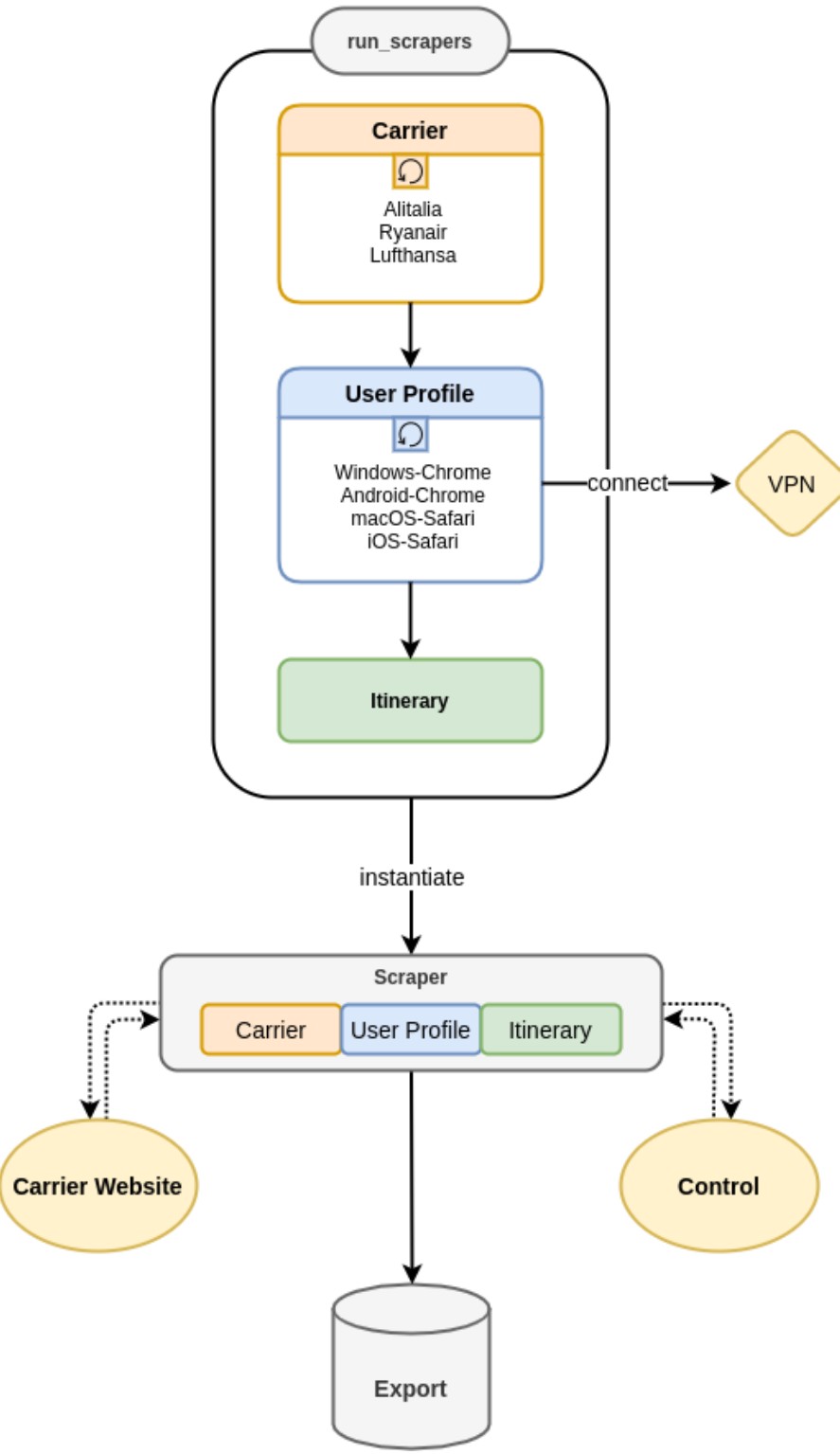

**Figure 2.** Software Architecture.

## 4. Results and Discussion

*4.1. Data Collection*

As described in Section 3 the research set out to collect data based on the following factors: (i) three carriers; (ii) four user profiles; (iii) four times a day; (iv) thirty day time span long time before the dates of the flights (4–5 months); (v) fourteen day time span near the dates of the flights (2–3 weeks) .

This consists up to a total of 1939 records, 1440 records (long time before the departure dates of the flights) and 672 records (near departure dates of the flights). Over the course of the research, some errors have been reported in the *logbook* file of the software during the data collection phase. Most of these errors resulted in the scrapers not collecting any data and are presented below. The final dataset at the end of the thirty days (long time before the flight) was composed of 1163 records. The final dataset at the end of the fourteen days (near the flight) was composed of 513 records. One of the most common errors occurred in the communication between the data acquisition software and the airline websites. The *logbook* contains errors, such as *NoSuchElementException* or *TimeoutException*, which indicate that some HTML elements were not loaded correctly. Whenever the scraper encountered this type of error, the scraping session for the respective user profile was aborted and no data was collected. This happened in 116 out of 237 reported error cases.

Seldom the Amadeus API failed to respond to the forwarded request. This meant not having control data for that particular session. As a result, the scraped data was not exported.

The data was collected using a client device and a scheduling script. On some occasions the client device was not available, which resulted in the scraper not being launched and the data not being collected.

*4.2. Data Cleaning*

Before inspecting the collected data, it was necessary to clean it to remove any of the issues that are presented below. After this process, the final record were 1132 (long time before flights) and 464 (near the flights).

Two types of errors originated from issues related to the VPN service. In one case, the connection to the server could not be established. In the other case, the requested server was not available and the VPN provider automatically redirected the connection to a different one. In both cases, the consequence was that the IP address seen by the airline website was not the intended one. For this reason, the collected data did not meet the requirements of the research anymore and was hence discarded.

In a few cases, it was noted that the total price scraped did not match the sum of the inbound and the outbound flight. In all the scenarios observed, the total price matched only one of the two. It has been concluded that during these sessions, the scraper had encountered issues in extracting the correct value. For this reason, the data collected from those scenarios has been rejected.

On a few occasions, the Ryanair control scraper was unable to finalize the scraping sessions. The collected data from these sessions was not taken into account.

The Lufthansa flights were affected by a schedule change on the 26 February. The departure flight LH1871 has suffered from a time change and the return flight LH1870 has been cancelled. In order to continue the data collection, the flight with the most similar characteristics (LH1872) was selected. Despite these schedule changes, there was no impact on the price and the fare collected from Lufthansa. Therefore, it has been decided that the data collected after the schedule change would still be valid for the research.

Subsequently, when the 14-days data was scraped, most of the original flights went through schedule changes or ran out of seats, due to the proximity to the departure dates. For this reason, other flights that had similar departure times were chosen. Also in this case, no evident price discrimination was observed. In a few instances, the scheduler was postponed to a later time by the operating system. As a consequence, the data was collected at a wrong time and had to be excluded from the analysis.

Between the two data collection periods, Ryanair went through an update of their user interface. The changes introduced affected the baggage selection for the single flights (departure and return). As a result, the sum of the single flight prices is not equal the total price scraped. This does not represent an issue however, because the prices of the single flights were only used as a way to check that the total price was correct. In light of this, when browsing through the dataset, one should keep in mind that the Ryanair prices scraped after the 25 June present this issue for the single departure and return flights, therefore the reader has to refer to the total price.

*4.3. Data Inspection*

The data inspection focused on determining price and fare differences across two axes. The first check was done between the scraped data using the cookies and the respective control group for the same user profile (intra user profile check). The second check was done between the scraped data using the four different user profiles (inter user profile check). Moreover, an analysis on differences in fare basis has been executed.

However, before describing these analyses, it is interesting to have a generic overview on the inspected data. Particularly, one can easily check that prices tend to increase as the distance from the departure date shortens. Precisely, the return price experiences a higher rise than the departure price. Thinking about the explanations of this evidence, at a first glance the scarcity argument seems to be quite plausible: near a departure, the price rises because of the shortage in available seats, and because customers tend to have a less elastic demand. However, the already examined literature has also proven how this argument is not always true since the price for a flight could also decrease near the departure according to many factors. Therefore, the increasing trend highlighted by our data has to be matched with the peculiar context during which this work has been done. In our case, the lower and stable prices in February and March can be reasonably explained as the result of the pandemic context, which makes consumers less prone to buy a ticket (because of uncertainty), and airline companies more willing to stimulate the demand. Thus, the context has pushed airline companies to incentive travelers to book a flight in advance by lowering prices. On the other hand, the higher prices in proximity of the departure and their intense volatility can be seen as a more common scenario: the context is much more predictable, and the airline companies can rise the prices for those travelers who have not bought the ticket in advance, trying to maximize their profits.

4.3.1. Intra User Profile Check

Turning back to the main focus of our analysis, it was first checked whether any price collected from the websites of the companies is different than the control price. For what concerns the long time before flights, out of the 1132 records available, 11 of them presented a difference in price.

Of these 11, only in 3 cases the scraped price was higher than the control price (see Table 3 and Figure 3).

**Table 3.** Price difference detected: website price higher than control.

| OS | Browser | Search Date | Search Time | Air Carrier | Website Price | Control Price | Seats Left |
|---|---|---|---|---|---|---|---|
| Windows 10 | Chrome 87 | 3 March 2021 | 09:01:14 | Alitalia | 73.88 | 73.88 | 2 |
| Android 10 | Chrome 87 | 3 March 2021 | 09:02:32 | Alitalia | 73.88 | 73.88 | 2 |
| Mac OS 10.15 | Safari 14.0 | 3 March 2021 | 09:03:51 | Alitalia | 73.88 | 73.88 | 2 |
| iOS 14.3 | Safari 14.0 | 3 March 2021 | 09:05:09 | Alitalia | 73.88 | 73.88 | 2 |
| Windows 10 | Chrome 87 | 3 March 2021 | 14:01:16 | Alitalia | 89.88 | 73.88 | 2 |
| Android 10 | Chrome 87 | 3 March 2021 | 14:02:35 | Alitalia | 89.88 | 73.88 | 2 |
| Mac OS 10.15 | Safari 14.0 | 3 March 2021 | 14:03:53 | Alitalia | 89.88 | 73.88 | 2 |
| Windows 10 | Chrome 87 | 3 March 2021 | 18:31:15 | Alitalia | 89.88 | 89.88 | 7 |
| Mac OS 10.15 | Safari 14.0 | 3 March 2021 | 18:33:32 | Alitalia | 89.88 | 89.88 | 7 |
| iOS 14.3 | Safari 14.0 | 3 March 2021 | 18:34:49 | Alitalia | 89.88 | 89.88 | 7 |

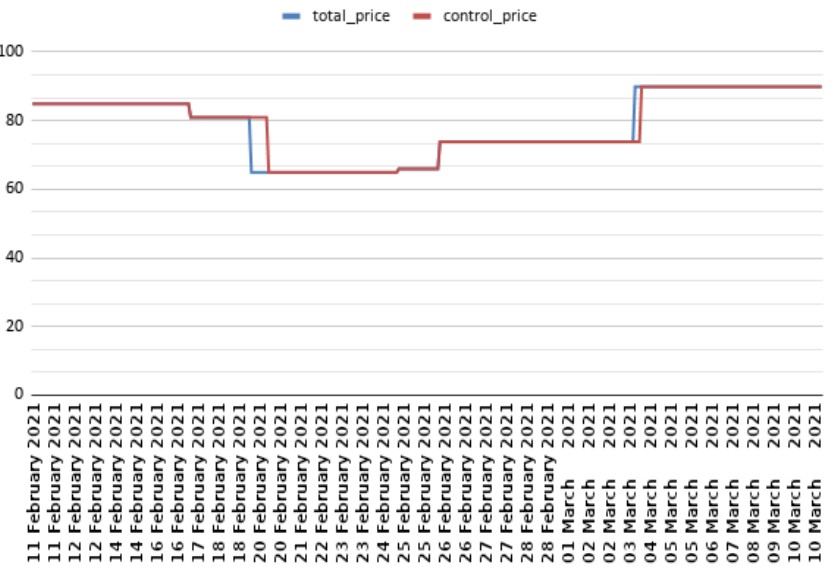

**Figure 3.** Alitalia price chart.

For the remaining 8, which all occurred on the 19th of February, the scraped price was lower than the control (see Table 4).

**Table 4.** Price difference detected: website price lower than control.

| OS | Browser | Search Date | Search Time | Air Carrier | Website Price | Control Price | Seats Left |
|---|---|---|---|---|---|---|---|
| Windows 10 | Chrome 87 | 18 February 2021 | 18:31:12 | Alitalia | 80.93 | 80.93 | 7 |
| Android 10 | Chrome 87 | 18 February 2021 | 18:32:30 | Alitalia | 80.93 | 80.93 | 7 |
| Mac OS 10.15 | Safari 14.0 | 18 February 2021 | 18:33:48 | Alitalia | 80.93 | 80.93 | 7 |
| iOS 14.3 | Safari 14.0 | 18 February 2021 | 18:35:04 | Alitalia | 80.93 | 80.93 | 7 |
| Windows 10 | Chrome 87 | 19 February 2021 | 09:01:13 | Alitalia | 64.93 | 80.93 | 7 |
| Android 10 | Chrome 87 | 19 February 2021 | 09:02:32 | Alitalia | 64.93 | 80.93 | 7 |
| Mac OS 10.15 | Safari 14.0 | 19 February 2021 | 09:03:51 | Alitalia | 64.93 | 80.93 | 7 |
| iOS 14.3 | Safari 14.0 | 19 February 2021 | 09:05:08 | Alitalia | 64.93 | 80.93 | 7 |
| Windows 10 | Chrome 87 | 19 February 2021 | 14:01:14 | Alitalia | 64.93 | 80.93 | 6 |
| Android 10 | Chrome 87 | 19 February 2021 | 14:02:31 | Alitalia | 64.93 | 80.93 | 6 |
| Mac OS 10.15 | Safari 14.0 | 19 February 2021 | 14:03:47 | Alitalia | 64.93 | 80.93 | 6 |
| iOS 14.3 | Safari 14.0 | 19 February 2021 | 14:05:07 | Alitalia | 64.93 | 80.93 | 6 |
| Windows 10 | Chrome 87 | 20 February 2021 | 09:01:15 | Alitalia | 64.93 | 64.93 | 5 |
| Android 10 | Chrome 87 | 20 February 2021 | 09:02:31 | Alitalia | 64.93 | 64.93 | 5 |
| Mac OS 10.15 | Safari 14.0 | 20 February 2021 | 09:03:47 | Alitalia | 64.93 | 64.93 | 5 |
| iOS 14.3 | Safari 14.0 | 20 February 2021 | 09:05:05 | Alitalia | 64.93 | 64.93 | 5 |

This could indicate a price discrimination in favour of customers, where the airline company offers a price discount to incentive sales and stay competitive. However, upon further analysis, it was noted that all 8 occurrences happened consecutively. This could indicate a delay in the GDS fetching the correct fares. This phenomenon is mentioned by [34] and it is referred to as *price caching*. Therefore price discrimination was excluded. Conversely, for what concerns the remaining three cases, where the scraped price was higher than the control price, all took place on the 3rd of March during the same scraping session for Alitalia (see Figure 3 and Table 3). In all 3 of them, the seats available were 2, which indicated that the fare bucket was about to increase. By further analysing the previous and the following scraping sessions, it was observed that the price difference could be due to a delay in the GDS fetching the correct fares. Again, this would rather point towards a caching of prices.

Another intra user profile check was done looking at the fare basis. The only difference that could be observed concerned Lufthansa. When executing the search directly on the airline website, the offered fare basis was *KEULGTX3* (hand-luggage fare from Lufthansa website). Whereas during the control search via the Amadeus API, the offered fare basis was *KEUCLSP3* (checked baggage fare from GDS). This was expected because of the limitations of the API described in Additional File S1 (see Supplementary Materials). To nullify this limitation and to make the two sets of data comparable, the letters *CLSP* were replaced with *LGTX* in the control data fare basis. By doing so, it could be concluded that no fare difference occurred.

For what concerns the observations near the flights, out of the 464 records available, 118 of them presents a difference in price (i.e., 25% of them). The 118 cases observed belong to Lufthansa exclusively, and they accounts for 72% of the total Lufthansa observations collected in the second span of time.

As it is possible to notice from Figure 4, the two prices only coincide during June 28th and 30th, July 1st and 2nd, and during the morning of June 27th. All these cases present one of the pairs of departure and return prices reported in Table 5.

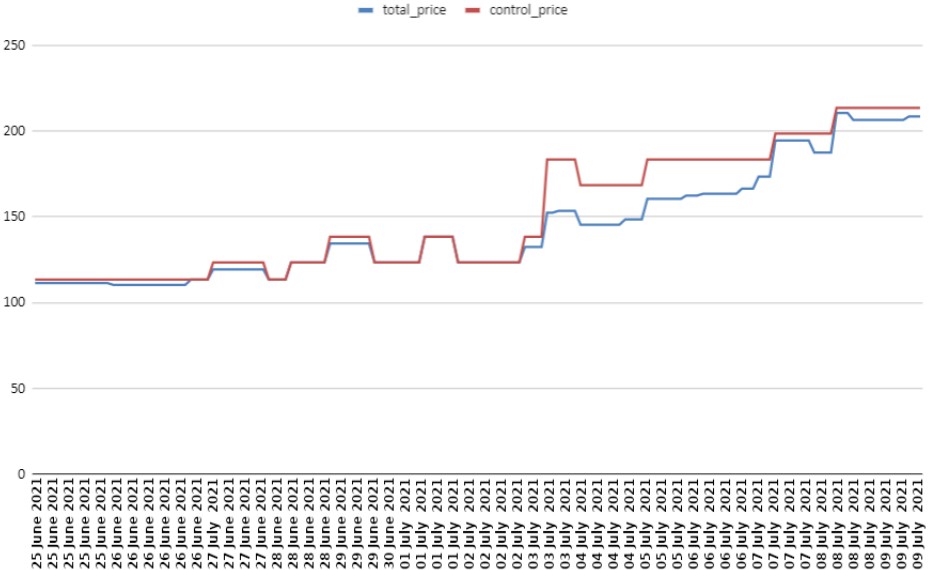

**Figure 4.** Lufthansa prices acquired near departure.

**Table 5.** Price pairs of Lufthansa flights.

| Departure Price | Return Price |
|---|---|
| 54.06 € | 59.27 € |
| 54.06 € | 69.27 € |
| 69.06 € | 69.27 € |

Moreover, the price directly collected from the website of Lufthansa is much more volatile than the control one. In particular, while the price of the website experiences 24 changes among 22 different prices in such a short range of time; the price collected from the GDS changes 13 times and it varies among 7 prices. Similarly, the difference between the price and its control is quite volatile. As shown by Figure 5, the price difference ranges from 2 € to 31 €, and it tends to increase through time. Though, it is not possible to discern a clear pattern.

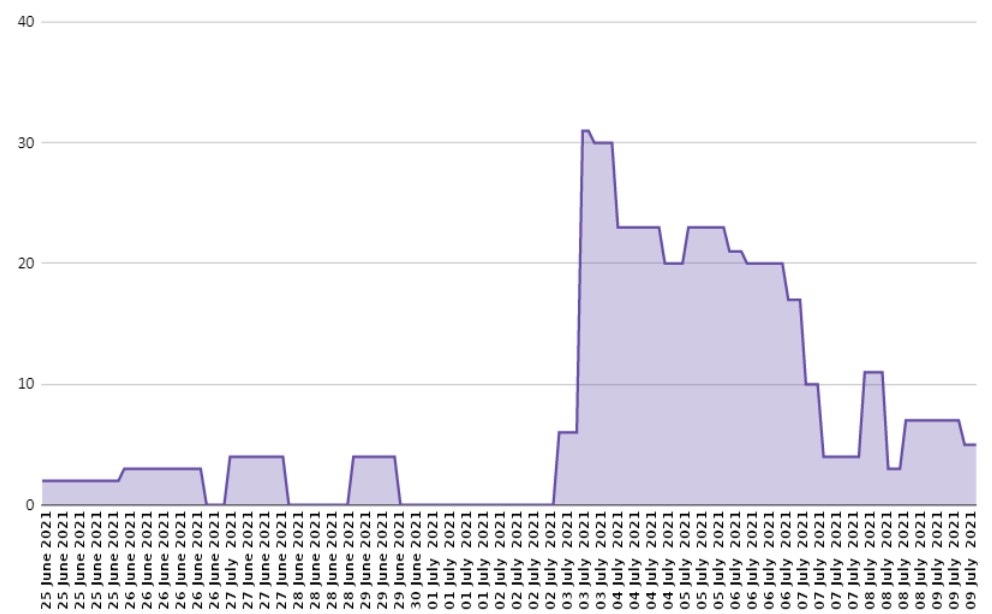

**Figure 5.** Price difference between Lufthansa prices and their controls.

However, a third conclusion can be drawn from Figure 4 and it is quite reassuring for consumers: in all the cases, the control price is higher than the collected one. Therefore, the airline company does not adopt discriminatory practices that negatively affects those travellers that decide to buy the ticket in the airline website. On the contrary, the airline company tends to reduce the price, if compared to the control one. If a discriminatory pricing practice is adopted, then it may not generate a price surcharge for the direct customer.

Nonetheless, it is important to determine the reasons behind this evidence. Indeed, if one looks at the already existing literature, once all the technical issues that may cause price discrepancies have been taken into account, then it is quite rare to still have to explain economically remarkable differences in price. Conversely, in our analysis, once we rule out technical issues—mainly ruling out volatility concerns through the control price, but also adjusting for price caching-, there are still relevant differences between the price displayed on the website and the one collected by the GDS. First of all, one has to take into account that the data are collected near to the departure, and so the airline company wants to sell all the seats left. In this scenario, it could be that, during our data acquisition period, Lufthansa used this channel to offer special discounts and lower prices to potential travellers so as to fill the plane. Moreover, competition could play a particularly relevant role. Indeed, it could be that this company tried to promote direct sales by offering lower prices and discounts so as to make the traveller less willing to buy from a middleman. The GDS price is a good benchmark in order to catch price volatility, but one has to take into account that it is not a final price (i.e., travelers do not directly buy from Amadeus). For this reason, it is possible that the discounts offered to customers are nothing but the results of competition among final sellers. In conclusion, one has to mention the pandemic scenario again. The necessity to recover from the remarkable economic losses could explainthis discrepancies between the website price and its control.

### 4.3.2. Inter User Profile Check

The second part of the analysis aims at checking whether personal information are used by airlines to set a different price to different customers for the same ticket. This is a particularly relevant analysis, since the only results that researchers have found when looking for price discrimination in online markets arise from differences in browser, operating system, geolocation, device, and so on. For this inspection, the prices analysed were

those coming from scraping the user profiles during the same sessions. The only price difference observed was in the Ryanair prices offered between desktop and mobile searches (see Table 6). Under deeper scrutiny, it was observed that in both cases the total price never matched the sum of the inbound and outbound flight. When comparing separately the departure and return prices, no price difference was observed between the user profiles. It was not possible to understand the reason behind the discrepancy between the desktop and mobile price. Nevertheless, it was noted that in almost all the cases the difference mounted up to €0.31, therefore too low to be considered price discrimination.

**Table 6.** Ryanair price discrepancy.

| OS | Browser | Air Carrier | Departure Price | Return Price | Discrepancy | Total Price |
|----|---------|-------------|-----------------|--------------|-------------|-------------|
| Windows 10 | Chrome 87 | Ryanair | 41.63 | 43.71 | 0.20 | 85.14 |
| Android 10 | Chrome 87 | Ryanair | 41.63 | 43.71 | 0.31 | 85.03 |
| macOs 10.15 | Safari 14.0 | Ryanair | 41.63 | 43.71 | 0.20 | 85.14 |
| iOs 14.3 | Safari 14.0 | Ryanair | 41.63 | 43.71 | 0.31 | 85.03 |

## 5. Conclusions and Future Directions

The aim of this work was to investigate the airline industry and the impact of digital technologies on it. Particularly, the focus of this work was on prices, since the enhanced ability of the supply side to obtain detailed information on the demand side could foster the adoption of personalisation techniques, such as price discrimation. In order to assess the presence of this practice, we designed and developed a software that is able to automatically search and acquire pricing data from the websites of the airlines. Additionally, the software exploits the API of the Global Distribution System Amadeus, the main flight booking platform dedicated to travel agents, in order to retrieve control data and to look at the potential differences in fare basis. We released both the data and the software publicly and we adopted the open-source paradigm. Over the course of six weeks, we acquired each day at four different time stamps pricing data of three different airlines (i.e., Alitalia, Ryanair, and Lufthansa) for one Italian and for one European itinerary to determine whether price discrimination occurs. Two acquisition were performed, the first one long time ahead the dates of the flights, the second one near the departure. Four distinct user profiles were identified to check whether there is a link between the device brand and type used to execute the search, and the offered price. We analysed the collected data by performing two types of inspection: intra-user profile and inter-user profile. The first one to ensure that no difference occurred among prices offered to the same user at different times with and without the presence of cookies. The second one to focus on the prices offered to each user profile defined by the operating system and browser.

The results from this research prove that the analysed airlines did not make use of price discrimination within the observed time periods. This holds true for both the inter-profile check where different user profiles were compared and the intra-profile check that focused on the potential impact of cookies on the search results. During the one-month and two weeks span of the research, none of the three airlines changed their offer depending on specific user characteristics. The ticket prices were consistently the same when comparing the data retrieved from the airline websites with the control data.

Our results derived from this empirical investigation are aligned with the already existing literature on this topic. It is true that airline companies have a direct channel to better understand the needs and preferences of each customer, but, in this study, they do not currently use these data to adopt price discrimination. What happens is quite the opposite, and the reason is to be found in the peculiar structure of this industry. Indeed, the ticket market is quite competitive, and a given seat in a flight can be booked by a traveller in many different ways: the ticket can be purchased directly from website of the airline; it can be bought by means of both an online or a brick-and-mortar travel agency; it can be found by means of a content aggregator; or it can be the result of a special deal

between the airline company and the company one works for. In a nutshell, the possibility of better segmenting the market thanks to the new granular information may have made the competition fiercer and the behavior of the firms work in favor of the demand side. The proof lies on the above mentioned current evolution of this market, and the way how the players are coping with new protocols, old data oligopolists, powerful search engines, and virtual platforms. On the one hand, airline companies want to better personalize the offer by managing not only flight seats but also rental cars, hotels, and all those services that surround a travel. On the other hand, they need to stay competitive by allowing indirect sales through the GDS - which were essential before the advent of digital technologies - and the OTAs to fence off competitors that would otherwise carve off parts of their market share. Consequently, travellers benefit from discounts, coupons, special offers, and highly differentiated products.

In discussing the results of this empirical work, it is also important to shed light on the potential limits of this work so as to pave the way for future research. In this work three European carriers have been observed and two routes were considered, a national one from Rome to Catania and a continental one from Rome to Munich. It would be very interesting to widen the scope and analyse the ticket prices of additional airline companies. Additionally, further research could look at different routes on the national, continental and intercontinental level. It could also be analysed whether the ticket prices might differ based on the geographical location of the user. Especially when looking at international flights, the research could be expanded towards multi-leg flights that include a layover between the departure and the arrival airport. In addition, the presented analysis was focused on leisure passengers. In the future, the research could be expanded onto business passengers to find out whether business class tickets are subject to price discrimination strategies. As presented in Section 2.3, there are several fare rules that have an impact on the ticket price. One of the fare rules presented is the advance purchase rule. It states that a user can be offered a specific price, if the search for the flight is done with a specific number of days prior to the intended date of travel. In the case of this work, the users were looking for flights on the second and the third weekend of July. Hence, the search was done with an advance purchase time of five months. Further research could include flight searches with different conditions. A user looking for a flight departing within the next 30 days would represent an interesting case because passengers could be more susceptible to price fluctuations close to departure. Moreover, the length of the stay could have an impact on the price. This research focused on users looking for a weekend getaway. Therefore, further analyses could take into account trips that include a longer period of stay at the destination. When it comes to user data, a possible analysis could include registered users that are logged into their profile versus users that search for a flight without identifying themselves. The aforementioned ideas to expand the research relate to the technical aspects of the setup.

Another field that could be of interest is that of consumer psychology. Many users claim that their browsing history has an effect on the content they see online. In certain ways, this is true, because companies can target people who have already visited their website with their ads. However, the findings of the presented research prove that this is not the case for airline ticket prices. Users whose browser had stored the cookie from the airline website were shown the exact same prices as users that came onto the website for the first time. If no proof of price discrimination based on cookies could be found, this raises the following question: why do users think that airlines raise their ticket price based on how often they have searched for a flight? One answer could be the phenomenon of illusory correlation where people perceive a direct relationship between two independent variables. For example, a user might have looked for a flight on Monday morning without booking it. When he searches for that same flight again later in the afternoon, the price is higher. The user might think that the ticket price has increased because this is already the second time he looks up the flight. However, there can be several other factors that might have caused the difference in price and of which the user is unaware. It would be

interesting to find out whether the debate on price discrimination is based on actual price differences or whether those differences are just perceived by users. A possible research setup could consist of people who actively look for flights on different airline websites and a scraper that automatically collects information on those same flights. After a period of time, the people would be asked how often the ticket prices have changed. Their answers would be compared to the data scraped directly from the websites to find out whether users might perceive differences in ticket prices more often than they actually occur.

A further aspect that could be considered in the future is that price discrimination does not necessarily result in different retail prices. In [26] the researchers analysed the search results of different e-commerce websites to measure price discrimination. They found out that the final price for a specific offer was the same between users. However, the order in which the results were presented differed heavily: Apple users who are generally expected to have a higher willingness-to-pay were shown the more expensive offers first, whereas Windows users were shown less expensive offers at the top of the page. This is known as price steering. More in-depth research could be done to analyse whether airline companies make use of the same strategy.

Finally, it must be kept in consideration that this research was done during the Covid-19 pandemic that has hit the airline industry in a particularly strong way. This historical phase has temporarily changed consumer behavior, especially in the airline industry. Travel restrictions, hygiene standards, and social distancing have led to an inevitable decline in the demand for air services and uncertainty about the medium-term outlook limits economic recovery for airlines. Furthermore, the sector remains exposed to a possible pandemic resurgence and national governments could impose new restrictions on air travel. This could threaten the existence of some companies in the sector, as production and revenues are likely to remain below pre-crisis levels for some time to come. According to the authors of [42] in 2020 air passenger transport decreased by 60 percent, compared to 2019 levels. The losses were 371 billion dollars, and the authors of [43] expect air traffic to return to 2019 levels only in 2024. In fact, most of the profits earned on a flight are generated by business travelers and related services, while leisure travelers are more sensitive to price changes and help cover fixed costs only. However, due to the pandemic, the number of passengers generating these profits has drastically reduced. Business travel will have a slower recovery, and in any case no more than about 80 percent of pre-crisis levels by 2024 [43]. In this context, remote working methods will also remain after pandemic crisis and company employees will tend to travel less for work. Conversely, leisure travel will tend to recover first when the pandemic is under control. Therefore, when the demand for air travel starts to grow again, it will probably exceed the initial supply. Airlines will have to organize themselves in time to restore delay management capabilities and this could lead to an increase in fares in the short term. In fact, many airlines will also face the cost of sanitary measures (e.g., disinfection, temperature controls or viral tests). Furthermore, social distancing measures could reduce the maximum occupancy of each flight by as much as 50 percent. In this context, as explained in [44], airlines could change their pricing policy, for example for long-haul flights. Today, most airlines have a higher price on direct flights. While business travelers book these non-stop flights, leisure travelers are more price sensitive and often choose a route with a stopover. Therefore, due to the decrease in business travel, it may be necessary to increase prices also for indirect flights. In conclusion, some airlines responded to the pandemic by restructuring themselves more efficiently, offering a personalized travel experience. Furthermore, investments in IT and automation will allow companies to simplify the check-in and boarding phases and the implementation of health protocols, in particular mobile apps will be used to archive vaccination certificates (green pass) and results of COVID-19 tests.

In this work possible effects of the pandemic on the pricing strategy of the airlines have been discussed in some extent. In order to further analyse whether airlines make use of price discrimination, similar analyses should be reproduced once the airline industry will have recovered from the current effects of the Covid-19 pandemic [45]. One thing to bear in

mind is that the post-Covid market may be substantially different from what we were used to. As shown in [46], several airlines are currently relying in part to government assistance. Despite this, numerous airlines, including large corporations like Latam and Avianca, had filing for bankruptcy, and some others are expected to follow. This could result in a market with fewer competition. In this scenario, we could see the implementation of new pricing strategies that researchers and economists are required to monitor.

**Supplementary Materials:** The following are available at https://www.mdpi.com/article/10.3390/jtaer16060126/s1, Additional File S1: PDF file that contains, additional information about the used tools and about the architecture of the software.

**Author Contributions:** S.A. and M.R. performed state-of-the-art and economic analysis, analysed data and interpreted the results. K.S. performed software implementation, data preparation, experimentation, analysis, and data interpretation. E.W. conceived the idea, designed, and directed research, supervised and validated software design, development, and the experimentation. S.A., M.R., K.S. and E.W. wrote the manuscript. All authors read and approved the final version of this manuscript.

**Funding:** This research received no external funding.

**Institutional Review Board Statement:** Not applicable.

**Informed Consent Statement:** Not applicable.

**Data Availability Statement:** The software and data are available at https://github.com/kevinsartiano/airline-price-discrimination-project, accessed on 6 September 2021.

**Acknowledgments:** The authors would like to thank Antonio Buttà for his precious advises and for supporting this work.

**Conflicts of Interest:** The authors declare no conflict of interest.

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
