# Peer review of "Price Discrimination in the Online Airline Market: An Empirical Study"

_jtaer, doi:10.3390/jtaer16060126_

Round 1

Reviewer 1 Report

The paper is interesting and novel, relevant for of practitioners, policy-makers and scholars, regarding the price discrimination in the online airline market, and it is my pleasure to review it.

The Literature and Methodology are sophisticated, systematic and comprehensive.

However, I would have some considerations and suggestions for improving the quality of the article.

The abstract is explicit, suggestive, but oversized. An abstract, beyond the informative aspects, must stimulate the reader's curiosity to continue reading the article, but not to provide a pleasing summary of the content. We recommend shortening the abstract to reasonable size.

The paper should have an Introduction section, distinct from the Literature, where (i) the subject and approach are configured; (ii) the topic is presented to attract the reader's interest, (iii) the own approach is positioned and some details of the specific research topic are given, and (iv) a general presentation of the structure of the paper is provided .

In the present form of the manuscript, the two sections seem to be merged, the authors suddenly go to theories, contributions, concepts, reviewing some research on this topic, etc. We recommend setting up a separate Introduction section, and then discussing the literature and previous studies in this matter.

Also, the authors should dedicate a short, explicit paragraph to research objectives, and, if the case, the working hypotheses. Consequently, in the final part of the paper, they should indicate how the paper achieved its objectives, validated or rejected the hypotheses, and to what extent the present results confirm or refute similar (or close) research on this topic. In other words, how does this contribution fit into the international flow of researches in the field.

The work is detailed and well argued. However, it abounds in technical details, phases of analysis ( detrimental for economic interpretations), which are difficult to follow and unattractive to understanding economic and commercial aspects, analysis of managerial decision etc. Often, reading fluency is interrupted by passages and screenshots (ex Listing 1,2,3: spreadsheet_tool.py) which, honestly, we do not see their usefulness and opportunity in context. They may be useful in attesting the authenticity of the data and analysis, and, only if the authors / editor consider their presence as absolutely necessary, we recommend placing them in the annexes, mentioning this in the body-text of the article, accordingly.

In fact, after this impressive, well-documented, but difficult and less fluent presentation, readers will look for additional economic and managerial explanations in the final part of discussions and Conclusions. The authors devote a generous space to these interpretations, but "send" them quite rapid the conclusions regarding the results and the usefulness of the present analysis, and they focus on the limitations of the study, and how these limitations could be addressed in future researches. Although useful and necessary in an academic paper, we consider that a better balance between discussions and conclusions of current research, and, respectively, limitations and further directions is necessary, especially to increase the significance, relevance and visibility of the article.

For example, an important element, which define the temporal context of the research, is the impact of the COVID-19 pandemic. It is honestly mentioned, but briefly, and is not debated, compared, etc., as would be expected in the case of this industry, obviously affected by this global phenomenon.

Thank you for the opportunity to review this paper and good luck!

Reviewer 2 Report

The article is interesting and deals with a topic that is of interest to researchers and the general public. Personally, I consider that there is little to comment except the length of the document (it is very extensive and detailed). When reading the article there are times when there are doubts whether the main objective is the analysis tool or the analysis. From the results section, it seems that the analysis is the main one and the tool developed complementary. If so, I would propose the following: transfer part of the content of section 3 (software architecture), and even some element of section 2, to Supplementary Material.

It is a suggestion to leave the article in a more reasonable length, without removing content.

Round 2

Reviewer 1 Report

Thank you for providing the revised version of your manuscript. My comments and suggestions were adressed, therefore, I can now endorse publication.